# Nanobiotechnology-Enabled mRNA Stabilization

**DOI:** 10.3390/pharmaceutics15020620

**Published:** 2023-02-12

**Authors:** He Xian, Yue Zhang, Chengzhong Yu, Yue Wang

**Affiliations:** Australian Institute for Bioengineering and Nanotechnology, The University of Queensland, Brisbane, QLD 4072, Australia

**Keywords:** mRNA, stability, nanobiotechnology

## Abstract

mRNA technology has attracted enormous interest due to its great therapeutic potential. Strategies that can stabilize fragile mRNA molecules are crucial for their widespread applications. There are numerous reviews on mRNA delivery, but few focus on the underlying causes of mRNA instability and how to tackle the instability issues. Herein, the recent progress in nanobiotechnology-enabled strategies for stabilizing mRNA and better delivery is reviewed. First, factors that destabilize mRNA are introduced. Second, nanobiotechnology-enabled strategies to stabilize mRNA molecules are reviewed, including molecular and nanotechnology approaches. The impact of formulation processing on mRNA stability and shelf-life, including freezing and lyophilization, are also briefly discussed. Lastly, our perspectives on challenges and future directions are presented. This review may provide useful guidelines for understanding the structure–function relationship and the rational design of nanobiotechnology for mRNA stability enhancement and mRNA technology development.

## 1. Introduction

Messenger RNA (mRNA) therapeutics, particularly mRNA vaccines, have witnessed great success during the COVID-19 pandemic [1]. Lipid nanoparticle (LNP) encapsulated-mRNA vaccines produced by Moderna (Spikevax^®^) [2] and Pfizer-BioNTech (Comirnaty^®^) [3] have helped millions of people against COVID-19. The success of mRNA vaccines could be attributed to the various merits of mRNA. First, mRNA does not enter into the nucleus [4], thus it will not insert into the host genome to cause unwanted risk compared to live-attenuated vaccines and DNA vaccines [5]. Second, the flexibility of the mRNA encoding sequence makes it possible to express multiple proteins with a single formulation [6,7]. Third, mRNA can be chemically synthesized in a few days, which is faster and more economic than conventional protein drugs produced in cells [8]. Lastly, compared to inactivated vaccines mainly stimulating humoral immunity, mRNA vaccines are more potent because they mimic the process of viral infection, generating both cellular and humoral immunity [9,10].

Despite numerous advantages, the use of mRNA in developing countries has been limited. The fragile structure of mRNA makes it susceptible to degradation, requiring strict shipping and storage conditions [11]. For example, the Pfizer-BioNTech vaccine can be stored under −80 °C for 9 months, while the Moderna COVID-19 vaccine can be kept under −20 °C for 6 months. Under 4 °C, the mRNA COVID-19 vaccines from Moderna and Pfizer-BioNTech have a shelf-life of 1 and 2.5 months, respectively [12,13]. Compared to double-stranded DNA, single-stranded mRNA is more prone to degradation [14]. As the final protein expression level correlates with the content of intact mRNA delivered into cells [15], it is of utmost importance to maintain mRNA integrity for its application.

Many factors destabilize mRNA during its delivery and/or storage. The major destabilizing factor is mRNA in-line hydrolysis. The oxygen of the 2′OH group of the ribose acts as a nucleophile and attacks the phosphate group, resulting in intra-strand breakage [16]. Base oxidation [13], non-optimized mRNA sequence [17], and high temperature [13] also contribute to mRNA instability (see details in Section 2). To mitigate mRNA degradation, mRNA molecules can be chemically modified or self-assembled into nanostructures to reduce in-line hydrolysis [18]. Yet the negatively charged surface of mRNA repulses with the anionic cell membranes, which hampers the effective internalization of mRNA into cells. Many nanomaterials have been developed (e.g., lipids [19] and polymers [20]) to improve mRNA delivery efficacy, including organ targeting [21], cellular uptake [22], and enhanced translation [23]. In these nanoformulations, mRNA is condensed inside nanomaterials via electrostatic and/or π-π stacking interactions, thus the stability against enzyme digestion and upon storage can be enhanced. Understanding the structurally dependent performance of nanomaterials for enhanced mRNA stability is crucial to the future design of functional nanomaterials in various mRNA applications.

As an important topic in mRNA technology, the stability of mRNA has been reviewed mainly from the aspect of molecular design [18]. Despite some reviews discussing nanomaterials for mRNA delivery [1,15,24,25,26,27,28], there are rare review articles focusing on how nanomaterials and/or nanostructures contribute to mRNA stability to the best of our knowledge. Herein, we firstly review nanobiotechnology-enabled mRNA stability. We begin by introducing the factors leading to mRNA instability, including in-line hydrolysis reaction and factors accelerating it, as well as some cellular degradation pathways. Then, strategies for enhancing mRNA stability are discussed, including the mRNA modifications that are usually incorporated, mRNA self-assembly, and nano-encapsulation, etc. We also briefly introduce the manufacturing considerations to achieve the long-term storage of formulations. In the last section, we identify the limits in current research and provide some insights for next-generation mRNA nanoformulation design.

## 2. Factors Contributing to the Degradation of mRNA

The typical mRNA structure includes five structural elements: one 5′ cap, one 3′ poly A tail, 5′ and 3′ untranslated regions (UTRs), and a protein-coding sequence (Figure 1a) [1,11]. The first four are regulatory elements. Although they are not eventually translated into proteins, they are essential to mRNA stability [1,11]. The degradation of mRNA is usually caused by in-line hydrolysis. In-line hydrolysis is the cleavage of a backbone phosphodiester bond in mRNA by a 2′ hydroxyl group of the ribose sugar that is in line with the phosphate group and leaves 5′ oxygen (detailed reaction mechanism shown in Figure 1b) [16]. As the name of reaction suggests, the two reagents for in-line hydrolysis are mRNA and water molecules. Bronsted acids and bases assist proton transfer of the hydrolysis reaction [16,29]. High temperature [30] and degradative enzymes (e.g., RNases) [29] accelerate the rate of mRNA in-line hydrolysis. In addition, highly oxidative species (e.g., oxidative metabolites) [31] oxidize mRNA bases inducing base changes such as guanosine to 8-oxoguanosine [18]. This reduces the peptide bond formation efficiency during translation, resulting in the accumulation of stacked ribosomes [18]. The stacked ribosomes then trigger mRNA degradation via the no-go decay pathway, where mRNA is cleaved at the site of stacked ribosomes via endonuclease-mediated hydrolysis [18,32].

The above events can occur either intracellularly or extracellularly [33]. Particularly, exoribonuclease and endoribonuclease may even degrade nanoparticle-packaged mRNA intracellularly and extracellularly [33,34,35]. While inside cells, unique cellular machineries such as decapping and deadenylation contribute to mRNA degradation [36]. For decapping, the 5′ phosphoester of the 5′ cap of mRNA (m^7^GpppG) is recognized and cleaved by decapping enzymes (DCPs) [37]. The activity of DCPs can be enhanced by the enhancers of decapping proteins (Edc1p and Edc2p) [38]. Inside cells, decapping and translation are balanced due to the competitive binding between the translation initiation factor (eIF4E) and DCPs to the 5′ cap [39]. Therefore, when mRNA translation is inhibited, mRNA is destabilized. Translation can be inhibited by either external factors such as viral infection and nutrient starvation [39], or by reducing the concentration of molecular machines used for mRNA translation, e.g., ribosomes and tRNAs [17,40]. It is noted that overcrowded ribosomes also led to ribosome queuing and ribosome collisions [41,42], which eventually results in mRNA no-go decay [32]. Hence, conditions such as ribosome levels should be optimized to balance the stability and translation of mRNA toward successful protein production [41,42].

For deadenylation, mRNA deadenylase complexes (e.g., Caf1–Ccr4 and Pan2–Pan3) cleave the 3′ poly A tail, leading to mRNA decay [43]. During this process, microRNA (miRNA) may complementarily bind to the 3′ UTR of mRNA via hydrogen bonding, blocking the movement of ribosomes, thereby inhibiting translation and priming mRNA degradation [1,4,5,11,13,27].

**Figure 1 pharmaceutics-15-00620-f001:**
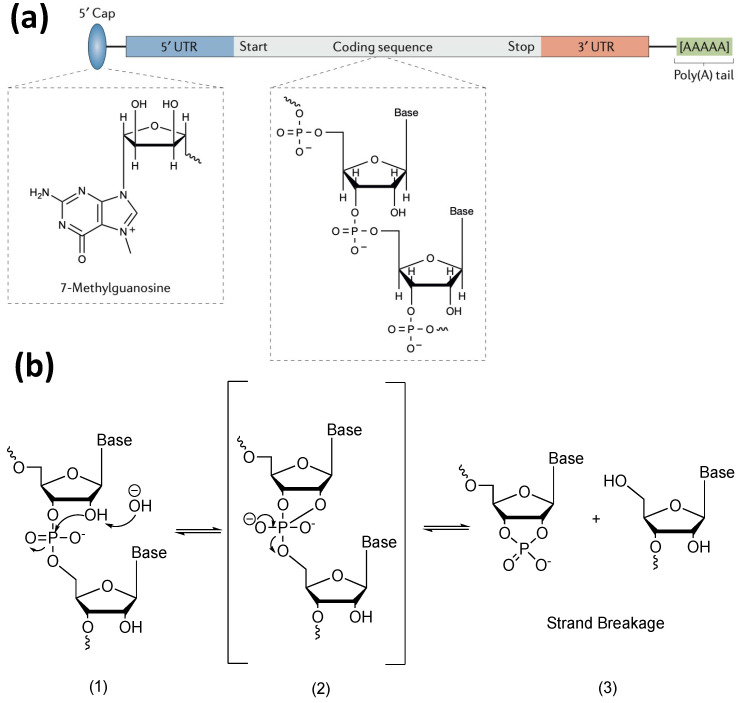
(**a**) Structure of native mRNA. Figure reproduced from Hajji and Whitehead (2017) with permission [44]; (**b**) the mechanism of base-catalyzed mRNA in-line hydrolysis. The base is represented by “OH^−^” in the figure. Mechanism reproduced from Pogocki and Schoneich (2000) via Chemdraw with permission [16].

## 3. Current Strategies for Improving mRNA Stability

Considering the fragile structure of mRNA, great research effort has been devoted to its protection to enable successful applications [1,11]. In this section, we review the mRNA structure design, its self-assembly, and mRNA encapsulated in nanoformulations, which are briefly summarized in Table 1. We discuss their structurally dependent performance with an emphasis on how these strategies counteract the destabilization factors. In addition, methods for enhancing the stability of mRNA in nanoformulations during storage are also included.

### 3.1. Molecular Design of mRNA

#### 3.1.1. Modification of mRNA Molecular Structure

The stability of mRNA can be enhanced by modification of mRNA molecular structures. The modification of mRNA molecular structure may occur either at the regulatory elements or the coding region. The regulatory elements (5′ cap, 3′ poly A tail, 5′ and 3′ UTRs) of mRNA have been modified for enhanced stability [1,4,5,11,13,27]. The standard 5′ cap can be replaced with analogs containing *O*-to-*S* substitution within the 5′ phosphoester. Due to this structural change, the recognition of mRNA caps by DCPs (e.g., Dcp2 and DcpS) is reduced, making mRNA more stable [45]. In the 5′ UTR, the first adenosine base can be modified to *N*^6^,2′-*O*-dimethyladenosine (m^6^Am), a large group that sterically blocks DCP access to the 5′ cap [18]. In the 5′ and 3′ UTR, base modification 5-methylcytidine (m^5^C) has been introduced to recruit human antigen R (HuR) protein binding, thereby prohibiting nuclease access [18]. The 3′ UTR sequences can also be adjusted to avoid miRNA complementary binding so that the movement of ribosomes during translation will not be blocked, therefore enhancing mRNA stability [46]. The length of the 3′ poly A tail is usually increased to counteract deadenylation that eventually results in mRNA decay [47]. In addition, removal of the 5′ and 3′ ends of mRNA that are necessary for exoribonuclease-mediated degradation forms circular RNA (circRNA), prolonging its half-life compared to linear RNA [48]. Moreover, incorporation of 5% *N*^6^-methyladenosine (m^6^A) into circRNA further enhanced its resistance to nucleases [49]. The readers are referred to some excellent reviews focusing on mRNA molecular structure and stability for detailed information [4,5,11,27].

In the coding region, both the mRNA backbone and nitrogenous bases can be modified. For mRNA backbone modification, the non-bridging oxygen atom can be replaced with sulfur through phosphorothioate modification (Figure 2a). As the S atom is less electronegative than O, mRNA with this modification becomes a bad substrate against RNase-mediated in-line hydrolysis [76]. Due to nucleophilic attack, the phosphorus atom becomes a chiral center that has two possible configurations: *S*p and *R*p. It is reported that the *S*p configuration of the central phosphate atom makes mRNA more stable compared to the *R*p counterpart, probably due to reduced RNase recognition [77].

For base modification in the coding region, the replacement of uridine (U) with pseudouridine (ψ) reduces mRNA in-line hydrolysis (experimental evidence shown in Figure 2b), which could be attributed to two reasons. One is that ψ downregulates 2′-5′-oligoadenylate synthetase activity, thus inhibiting the activity of 2′-5′ oligoadenylate synthetase-dependent ribonuclease (RNase L) [78,79] and reduces RNase L-mediated in-line hydrolysis. The other is that the nucleophilicity of the 2′OH group of the modified nucleotide is reduced [36]. This molecular modification strategy has been successfully applied in some mRNA cancer vaccines and COVID-19 vaccines [80].

#### 3.1.2. Construction of mRNA Nanoassemblies

The construction of mRNA nanoassemblies (3D nanostructure) is another strategy to enhance mRNA stability as there are many enzyme cleavage sites exposed in the linear stranded mRNA [50,51]. The mRNA nanoassemblies can be categorized into two types: intramolecular assembly within a single mRNA strand, and intermolecular assembly of multiple mRNA molecules. In some cases, both the intramolecular and intermolecular assemblies exist in one mRNA nanoassembly.

For the intramolecular assembly, its sequence (primary structure) needs to be carefully designed. Nucleobases, e.g., guanosine (G) and cytosine (C) can form three hydrogen bonds between each other while there are only two hydrogen bonds between adenosine (A) and uracil (U). Therefore, higher GC content is expected to enhance intrastrand hydrogen bond formation for the formation of mRNA secondary structure (e.g., hairpin loops) [50].

For the intermolecular assembly approach, researchers transcribed plasmid DNA containing spacer sequence (black) and G-quadruplex (G4) (yellow) motif (G_X_-N_1–7_-G_X_-N_1–7_-G_X_-N_1–7_-G_X_, (x = 3 to 6 and N (A, G, C or U)) into multiple long RNA strands [52]. As shown in Figure 3a, G4 motifs from different strands cross-linked via hydrogen bonds, and cations intercalated into the space of those cross-linked RNA via multivalent electrostatic interactions, further stabilizing the RNA. This secondary structured association allows the assembly of RNA-G4 into 3D hydrogel [52]. The viscous nature of hydrogel makes mRNA resistant to RNase access, thus prolonging the half-life of RNA-G4 (RG4) in serum compared to linear one (RGx): 50 vs. 35 h (Figure 3b) [52]. Another strategy is to hybridize complementary mRNA strands for enhanced mRNA stability. Compared to single-stranded mRNA, the stability of hybridized mRNA improved by 3000-fold against RNase A degradation [53]. The fancy of mRNA strand hybridization could also be applied in designing oligonucleotide linkers that join multiple single-stranded mRNA molecules together, driving the formation of 3D mRNA nanoassemblies (Figure 3c, left image). The prepared nanoassemblies hinder the RNase access due to steric hindrance which significantly reduced serum degradation compared to the mRNA without linkers (Figure 3c, right image) [54]. In both intermolecular approaches, only one type of mRNA was used as precursors. To our knowledge, rarely, reports used various types of mRNA precursors and assembled them into a single formulation.

### 3.2. Encapsulation of mRNA via Nanomaterials

Although stable mRNA nanoassemblies can be designed, their negative surface charge hinders cellular uptake due to membrane repulsion. Nanomaterials are used to address this challenge [15]. Nanomaterials-mediated mRNA delivery is either via in situ self-assembly into three-dimensional nanostructures or loading on pre-synthesized nanoparticles (NPs).

#### 3.2.1. LNPs

LNPs are the most widely used nanovectors for mRNA delivery to date [1,81], shielding the access of RNases to mRNA. LNPs are usually composed of four components: cationic/ionizable lipids, helper lipids (e.g., phospholipids), poly(ethylene glycol) (PEG)-modified lipids (a.k.a. PEGylated lipids), and cholesterol [1,44,81]. The suggested structures of mRNA-encapsulated four-component LNP are shown in Figure 4a [81]. The cationic/ionizable lipids are responsible for mRNA binding. Helper lipids such as phospholipids improve targeted cellular uptake. Poly(ethylene glycol)-modified lipids (i.e., PEGylated lipids) and cholesterol are responsible for mRNA-LNP stability [1,81]. In an earlier study, researchers coated mRNA onto a positively charged liposome (forming RNA-lipoplex) at a charge ratio [lipid (+): mRNA (−)] of 1.3:2 [82]. This made mRNA fully resistant to mouse serum degradation, thereby effectively transfecting the spleen upon injecting RNA-lipoplex intravenously [82]. Current research has moved on to tuning the chemical structure of lipids to control the stability of LNPs for controlled mRNA release and reduced degradation.

The cationic/ionizable lipids usually include a hydrophobic tail and a cationic or ionizable head (e.g., primary, secondary, and tertiary amine that can be protonated into cationic groups) [1], which have been used to condense mRNA and enhance its stability. Recently, cationic lipid head groups has been tuned with imidazole groups (Figure 4b) to provide π–π stacking for enhanced layer stability and higher protein expression after storage under 4 °C for 25 weeks than commercial controls (Figure 4b) [75]. It is suggested that the enhanced expression could also be attributed to the π–π stacking interaction between mRNA nucleobases and imidazole, as well as the antioxidant property offered by imidazole [75]. It is also reported that the imidazole group can promote endosomal escape [84], which can also increase the level of protein expression, suggesting the possible multi-functions of imidazole modification. In addition to lipid head modification, the concept of π–π stacking stabilized layer has been applied in lipid tail modification. Zhang et al. incorporated a benzene ring in the alkyl chain of ionizable amine (benzylamine) to stabilize the layer (Figure 4c) [83]. Compared to LNPs using unmodified lipids, those containing benzyl amines in the lipid tails exhibited enhanced colloidal stability (stable even after 135 days at 5 °C under serum treatment), and five-fold higher expression level [83]. For other LNP components (phospholipid, PEGylated lipid, and cholesterol), there are reports on tuned compositions for organ targeting and endosomal escape etc., but rarely on stability [85,86,87].

Ball and coworkers reported an interesting strategy to improve the efficacy of LNP delivery systems [88]. When mRNA and siRNA were co-delivered by LNP, or one of the RNA was replaced by an anionic polymer (e.g., polystyrenesulfonate (PSS)), the delivery performance could be improved. For example, compared to mRNA-LNPs without PSS, the co-delivery of PSS and mRNA led to significantly higher luciferase expression level [88]. The authors attributed the possible reason to the additional negative charge of the “helper polymer” (either one of the RNA or PSS, which could increase the electrostatic attraction inside the particle and promote the formation of a more stable and/or compact LNP) [88]. This strategy is useful for co-delivery, also has clinical relevance and significance by reducing required dosage of RNA.

#### 3.2.2. Polymer-Based NPs

Polymer is another class of materials for mRNA delivery, protecting mRNA via steric hindrance of anionic protein against mRNA [89]. A conventional method for mRNA delivery is achieved by the assembly of mRNA molecules and cationic polymers. Polyethyleneimine (PEI) is a conventional cationic polymer to load mRNA. However, its high charge density leads to a high N/P ratio when loaded with mRNA, making the polyplex surface positively charged [28,55]. This results in the adsorption of negatively charged molecules (e.g., serum proteins), leading to mRNA degradation.

One way to resolve above issue is to design branched polymers with lower charge density compared to PEI, e.g., single branch poly(*β*-amino ester) (PBAE). The linear portion of PBAE is for the electrostatic condensation of mRNA, and the branched part is for shielding of anionic molecules (Figure 5a) [56]. As suggested by the gel electrophoresis result, the hyperbranched PBAE retarded the movement of mRNA at a ratio of 5:1, while that of its linear counterpart was only 10:1 [56]. Polymers with positively charged side chains derived from natural amino acids such as poly(l-ornithine) and poly(l-lysine) have also been applied in polymer design. It was found that complexation using poly(l-ornithine) with a trimethylene spacer better enhanced mRNA from RNase attack than poly(l-lysine) possessing a tetramethylene spacer [59]. It was postulated that the difference in one methylene unit difference affected the ion pairing between mRNA phosphate group and amino groups in the positively charged side chain [59].

Moreover, the polyplex surface charge can be controlled to avoid the high positive charge density. Binary polyplexes (e.g., mRNA + poly(l-ornithine) mentioned above) can be further coated with negatively charged polymers to offer mRNA protection against serum [59,60]. This given function is similar to that of zwitterionic polymers. Zwitterionic polymer is designed with one positively charged group to complex genes inside and one negatively charged group, facing outside, repelling serum proteins [61,62]. However, those zwitterionic polymers require to be directly synthesized from limited types of monomers so that the chemical diversity of resulting polymers is limited [57,63]. To circumvent this issue, Liu et al. used a post-modification strategy—phospholipidation, to form zwitterions (Figure 5b). The phospholipid part led to hydrophobicity, together with the negative charge offered by the phospholipid head, enhanced protein expression in vivo compared to unmodified zwitterionic polymers (ten-fold higher) [57]. The authors attributed the enhanced expression to the enhanced cellular uptake/endosomal escape due to the similar nature of phospholipidated polyplex surface to biomembranes [57].

Another strategy is the combination of cationic polymer and cationic polymers modified with a neutral polymer to form core-shell structures. Examples of cationic polymers are PEI [64] and poly(amino-*co*-ester)s (PACEs) [65]. An example of a neutral polymer is PEG. In the study conducted by Grun et al., PACE and PACE-PEG condensed mRNA via electrostatic interaction and self-assembled into micelle with PEG facing outwards (Figure 5c) [58]. The PEG portion could sterically shield the core to prevent enzyme degradation and prevent particle aggregation. Formulations with 1% PACE-PEG content remained stable under sodium acetate buffer for 72 h, while those formulations with less PACE-PEG content only remained stable for 8 h [58].

Furthermore, functional chemical motifs can be introduced to above co-polymer systems to enhance mRNA stability. For example, encapsulating mRNA into a non-crosslinked polyplex micelle of PEG-poly(L-lysine) efficiently protected the remaining mRNA by more than 10,000-fold compared to naked mRNA in serum [33]. Moreover, cationic charge-preserved disulfide crosslinking (e.g., 1-amidine-3-mercaptoproyl group) improved mRNA stability to 4.9-fold compared to non-crosslinked polyplex micelle [33]. RNA oligonucleotides were also fabricated to link mRNA and polycation segment of PEG-block copolymers, driving the formation of polyplex micelle, further protecting mRNA from RNase attack [90]. Encapsulation of mRNA with two PEG-polycation block copolymers conjugated with phenylboronic acid or polyol groups allows spontaneous phenyl-boronate ester formation, driving the formation of a crosslinked polyplex micelle. This significantly improved the resistance of mRNA against RNase [60]. The additional component(s) of the copolymer system also endows sufficient mRNA release in the cytosol, priming enhanced translation.

Different from above strategies, the stability of mRNA under various pH can be controlled by the odd or even number of cationic groups. Uchida and coworkers reported that mRNA condensed from odd-numbered aminoethylene repeats remarkably protected mRNA from RNase degradation than that from even-numbered ones, leading to enhanced protein production [66]. The authors attributed the reason to the difference in the protonation of terminal primary side chain amino groups (pH 7.4), but the exact biochemical mechanism remained to be explored further [66].

Collectively, all the above studies use polymers to provide physical protection against anionic proteins (e.g., RNases) to enhance mRNA stability. Inspired by intracellular biochemical degradation pathways, Li et al. preassembled the translation initiation complex using polyamine, mRNA, and the negatively charged eukaryotic initiation factor 4E (eIF4E) (Figure 5d) [23]. This polyamine promoted eIF4E binding to the m^7^G-cap of mRNA, thus protecting the mRNA 5′ cap from DCP attack and enhancing mRNA stability. Compared to mRNA alone, the expression level of luciferase by preassembled mRNA is two-fold higher. Moreover, the authors also suggested the assembled polyplex provided a rigid shell to reduce the access of RNases [23]. Inspired by the design of branched cationic polymers, the translation initiation complex can also be designed by using branched polymers as scaffolds to provide extra protection.

#### 3.2.3. Hybrid and Other Strategies

Hybrid strategies for mRNA delivery are often referred to as the combined use of lipids and polymers [24,25]. As discussed before, while a higher N/P ratio leads to better condensation of mRNA using cationic polymers, it also suffers from positive surface charge-induced anionic displacement [55]. Formulating the polymer-condensed mRNA into LNPs could resolve above issue via steric hindrance and enhance biocompatibility [91]. Kaczmarek et al. mixed the PBAE-condensed mRNA with PEGylated lipids. The expression level of this hybrid formulation was two-fold higher than its counterpart without PEGylated lipids (Figure 6a) [67]. In addition, it was also suggested that polymers at the LNP core may further prevent the leakage of mRNA probably due to enhanced electrostatic interactions. As demonstrated by Islam and coworkers, increasing proportion of cationic polymer G0-C14 (with a cationic center G0 and four aliphatic tails C14) to mRNA in an LNP formulation successfully reduced mRNA leakage under serum treatment [22].

The strategies mentioned above (cation polymer-loaded mRNA encapsulated in LNP core) mainly address the limitation of cationic polymers. Although LNPs can successfully protect mRNA from RNases, the mRNA-LNP stability (aggregation) under various delivery conditions remains problematic. Polysaccharides and acid polymers can form a viscous hydrogel network with nearly neutral pH. Because neutral environment hinders the formation of Bronsted acids and bases, mRNA in-line hydrolysis is reduced. The aggregation of mRNA-LNPs loaded into hydrogel is minimized as the hydrogel is viscous (Figure 6b). For example, hyaluronan hydrogel loaded with mRNA-LNPs retained mRNA stability for 14 days under room temperature storage compared to free-standing mRNA-LNP (3 days) evidenced by the Western blot analysis [68]. Hydrogels also have potential in maintaining stability under physiological conditions. For example, hydrogels formed by chitosan remained stable (no swelling) under body fluid from tumor microenvironment treatment (Kras pancreatic cell-conditioned medium and leukemia cells in mouse macrophage-conditioned medium) for 28 days, making them a stable carrier for sustained mRNA-LNP release [69].

In fact, hydrogels can also be designed to directly load naked mRNA. For example, graphene oxide (GO) and low-density PEI can be used to fabricate injectable mRNA hydrogel. mRNA is mixed with PEI and probably protected by the π–π stacking interaction between the six-member rings of GO. This encapsulated mRNA exhibited better stability compared to the naked ones (48 h vs. 6 h) [70].

### 3.3. Other Considerations before Manufacturing

The mRNA nanoformulations need to be frozen or lyophilized before shipping [71,92,93]. Both freezing and lyophilization require deep freezing at −90 °C to −60 °C to reduce the rate of in-line hydrolysis [30].

For freezing, samples are required to be stored at ultralow temperature storage to stop most chemical reactions [71]. For example, the Pfizer-BioNTech and Moderna COVID-19 vaccines can be stored under −90 °C for 90 days, and the Moderna vaccine can be stored at −15 °C to −20 °C for 6 months [27]. The strict storage requirements on frozen mRNA formulations call for other solutions.

Lyophilization is developed to address above limits. The storage condition is milder for lyophilized mRNA nanoformulations, e.g., under refrigerator temperature. Lyophilization requires two additional steps to freezing: primary drying (sublimation) and secondary drying (desorption). The ultimate goal of these two processes is to remove water molecules from the mRNA formulations to prohibit in-line hydrolysis and maintain particle size [16,29]. Muramatsu et al. developed a three-step lyophilization process, including a freezing step at −45 °C, a primary drying step at −25 °C 26.6 mBar, and a secondary drying step at 30 °C 26.6 mBar. The obtained mRNA-LNPs remained stable at 4 °C for 24 weeks confirmed by DLS and protein expression results compared to non-lyophilized samples [72].

However, mRNA formulations may be damaged in the deep freezing and thawing process during freezing or lyophilization [71,92,93]. This is because there is a hydration layer formed by water molecules surrounding mRNA molecules and their nanoformulations (e.g., via hydrogen bonds). During above processes (transition from liquid water to ice and vice versa), the movement of water molecules disrupts the hydration layer. This poses mechanical stress onto the mRNA-LNPs, resulting in the aggregation of mRNA nanoformulations. Therefore, cryoprotectants such as saccharides are introduced [73]. The addition of sugar cryoprotectants lowers the concentration of water, therefore inhibiting ice development [28,74]. In addition, sugar cryoprotectants can interact with the NP surface (e.g., phospholipid heads of LNPs) via hydrogen bonds [28,74], displacing the water hydration layer on the NP surface. Thus, the movement of water molecules during freezing and thawing has minimal effect on the shape and size of the whole mRNA formulation, minimizing mRNA leakage, and thereby enhancing its stability [71]. After liquid nitrogen storage for 3 months, formulations with cryoprotectants maintained their size, while those without cryoprotectants aggregated [71].

## 4. Conclusions and Outlook

As an intermediate product of conveying genetic information, mRNA plays an indispensable role in every cell of life. mRNA therapeutics have attracted great research interest due to the intrinsic versatility of mRNA. The development of nanobiotechnology allows the successful delivery of mRNA to target cells, yet stability is a major concern. In our review, we first discussed mechanisms of mRNA degradation, followed by mRNA molecular designs including sequence modification and nanoassemblies, and various nanomaterials for mRNA delivery with design principles of addressing instability origins. Moreover, considerations during formulation manufacturing process are also briefly discussed. Despite current progress, there are still challenges remaining to provide a better understanding of the nanobiotechnology-enabled mRNA protection. Some questions to be answered are listed below.

(1)The biological impact of mRNA with molecular designs is to be evaluated. Although mRNA sequence can be sophisticatedly designed with enhanced stability, comprehensive studies of the bio-distribution and bio-safety evaluations of such exogenous mRNA in vivo or in clinical studies are needed.(2)The contribution of nanomaterial-enhanced mRNA delivery or stability should be investigated and elucidated. Considering the multiple steps in mRNA delivery, e.g., organ accumulation, cell targeting, endosomal escape, and protein translation, tuning the nanocomposition/structure could also impact those delivery steps in addition to interaction/protection with mRNA molecules. To date, current research generally evaluated the final protein expression level as the indicator to evaluate the so-called protection effect or stability, which may not be accurate. For example, an mRNA molecule may be stable in its structure, but has low efficacy in translation into proteins. Using biotechnology methods such as polymerase chain reaction (PCR) may provide mRNA sequence/structure information to better understand the stability change in various delivery steps and contribution to final enhancement in protein expression.

We also see more opportunities in the advanced nanobiotechnology-enabled mRNA stability.

(1)To tackle base mutation generated by oxidation. For example, in cancer microenvironment, high reactive oxygen species may be a factor leading to mRNA mutation and hindering mRNA function [94]. To date, there are rare reports integrating antioxidant molecules to alleviate the oxidative stress in those mRNA formulations for better protection, which is worth of future study.(2)To tackle RNases-mediated in-line hydrolysis. Allosteric regulators may be considered to inhibit RNase activity in a nanoformulation [95].(3)To tackle intracellular mRNA decapping. As discussed in Section 2, mRNA 5′ decapping can be counterbalanced by the binding of translation initiation factors. Despite mixing cationic polymers, translation initiation protein and mRNA have shown to enhance mRNA stability [23]; this strategy is limited to one type of translation initiation protein. It is expected that activating the cellular pathway (mechanistic target of rapamycin complex 1 (mTORC1)) responsible for recruiting multiple translation initiation factors, e.g., eIF4E and eIF4G [15] may provide better performance. Glutamine is a reported biomolecule to activate the mTORC1 [96]. It is expected that incorporating glutamine into nanomaterials may enhance mRNA stability.(4)For multi-stranded mRNA self-assembly, current building blocks are generally limited to mRNA expressing one type of protein. Future study on design and assembly of mRNA expressing different proteins may be of interest to applications such as multivalent vaccine application.(5)For lipid-based mRNA formulations, future effort may be devoted to the enhancement of mRNA-LNP structural stability as it prevents mRNA leakage before arriving at the targets. Inspired by current research on control over cationic lipids for tuned surface charge etc., other lipid components such as helper lipids and PEG may be tuned to control the shell structure of lipid for better colloidal and mRNA stability.

## Figures and Tables

**Figure 2 pharmaceutics-15-00620-f002:**
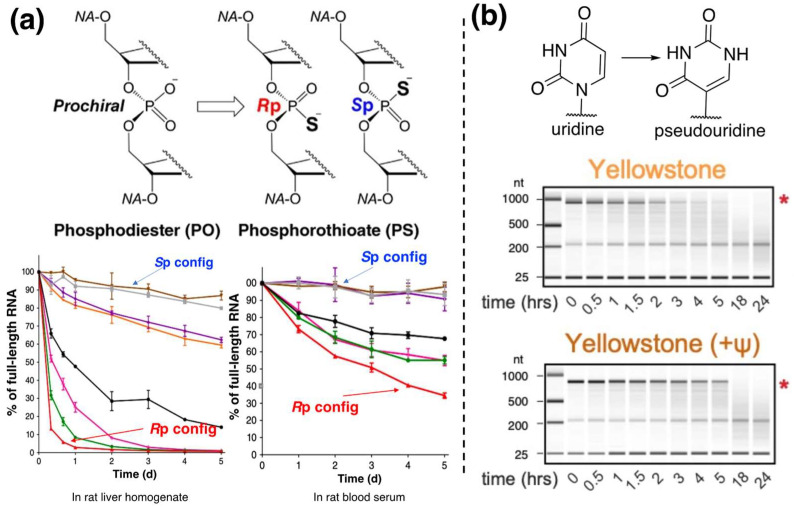
The effect of mRNA chemical modifications of on its stability. (**a**) Phosphorothioate modification and the effect of two configurations of the phosphorous atom on mRNA stability in rat liver homogenate and rat blood serum. Reproduced from Butler et al. with permission [77]; (**b**) evidence of pserudouridine modification on mRNA stability (confirmed by gel electrophoresis), * means more/less degradation. Reproduced from Leppek et al. with permission [36].

**Figure 3 pharmaceutics-15-00620-f003:**
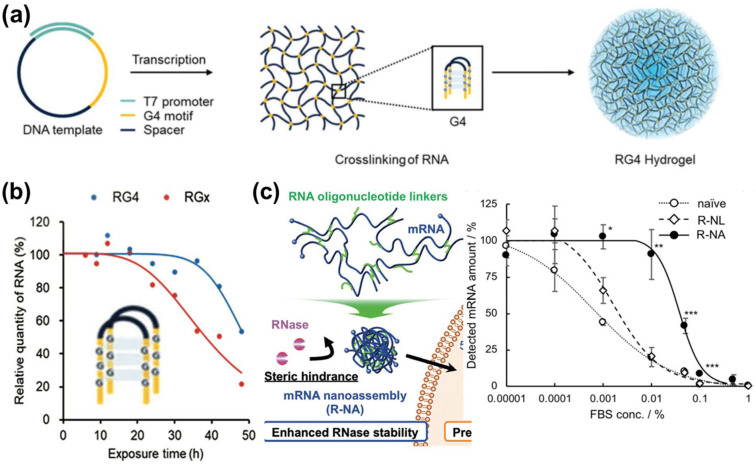
Construction of mRNA-NPs via manipulation of its sequence and evaluation of their stability; (**a**) scheme of the synthesis of free-standing protein-encoding hydrogel, reproduced with permission [52]; (**b**) relative quantity of RNA-G4 in blood serum, reproduced with permission [52]; (**c**) construction of linker-driven mRNA nanoassemblies (R-NA), their detected amount in FBS of various concentrations, and their remaining amount in mouse brain 4 h after injection. Reproduced with permission [54]. * *p* < 0.05, ** *p* < 0.01, *** *p* < 0.001.

**Figure 4 pharmaceutics-15-00620-f004:**
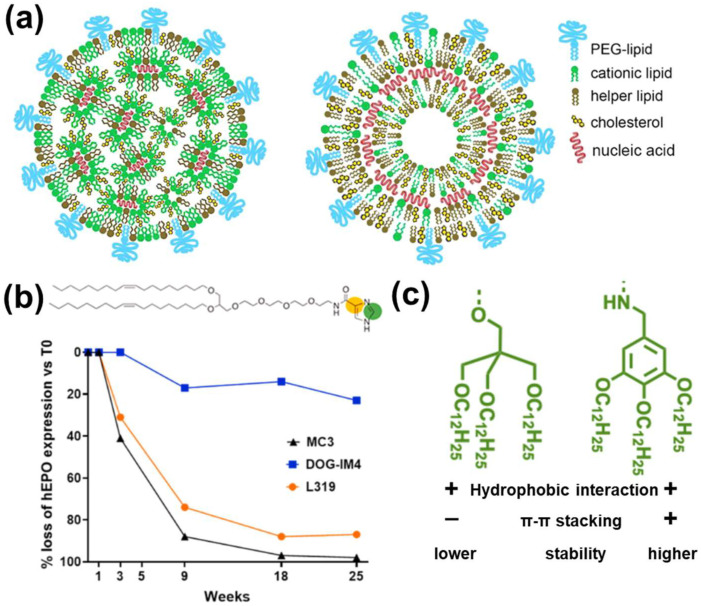
(**a**) Two suggested structures of mRNA-encapsulated LNPs, reproduced with permission [81]; (**b**) the loss of expression of mRNA encapsulated in LNPs containing imidazole (DOG-IM4), and tertiary amines (MC3 and L319) as ionizable lipids, reproduced with permission [75]; (**c**) a structural comparison between ionizable lipids with and without aromatic groups in the lipid tail, reproduced with permission [83].

**Figure 5 pharmaceutics-15-00620-f005:**
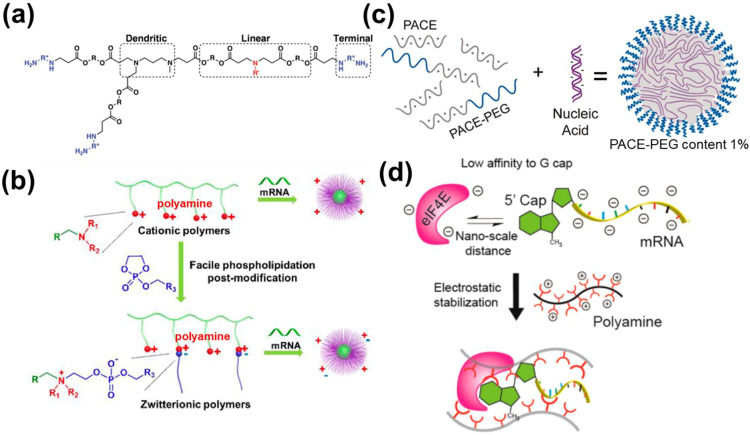
Polymers and mRNA stability. (**a**) The structure of a branched PBAE, reproduced with permission [56]; (**b**) phospholipidation of cationic polymers creates zwitterionic polymers, conferring higher stability against negatively charged proteins, reproduced with permission [57]; (**c**) the loading of nucleic acids on conjugated polymer PACE–PEG and formation of micellar structure. Reproduced with permission [58]; (**d**) preassembly of mRNA translation initiation complex prohibits decapping, reproduced with permission [23].

**Figure 6 pharmaceutics-15-00620-f006:**
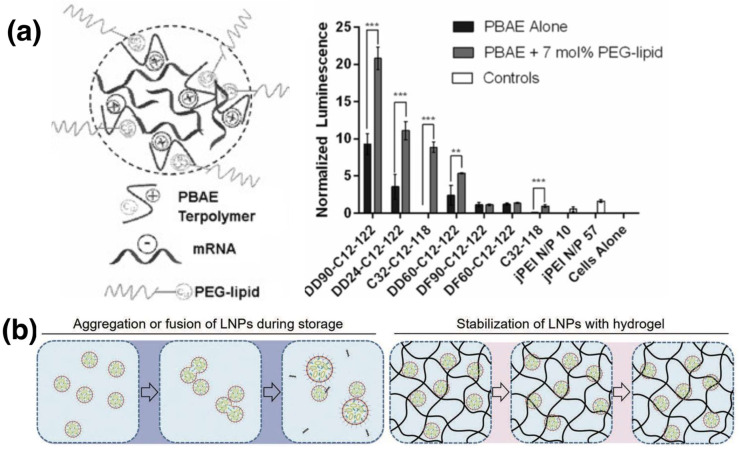
Schematic illustration of polymer-lipid hybrid strategies. (**a**) Polymer–lipid nanoparticles formed with PBAE and PEG-lipid with their stability verified by luminescence level, reproduced with permission [67]; (**b**) stabilization mechanism of mRNA–LNP injected in hyaluronan hydrogel [68]. ** *p* < 0.01, *** *p* < 0.001.

**Table 1 pharmaceutics-15-00620-t001:** Factors leading to mRNA degradation and brief description on their counteracting strategies.

Factors Destabilizing mRNA	Possible Counteracting Strategies	Reference
RNase-mediated in-line hydrolysis	(1) Modification of mRNA regulatory elements, nucleobases, and backbone; (2) Optimization of mRNA sequences; (3) Loading/encapsulation of mRNA on/into nanoparticles	[4,5,11,13,18,27,45,46,47,48,49] [50,51,52,53,54] [22,23,28,33,55,56,57,58,59,60,61,62,63,64,65,66,67,68,69,70]
High temperature	Freezing or lyophilization of nanoformualtions with the addition of cryoprotectants	[16,29,30,71,72,73,74]
Oxidative species	Introducing chemical groups (e.g., imidazole) with antioxidant properties	[75]

## Data Availability

Not applicable.

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
