# Peer review of "Nanobiotechnology-Enabled mRNA Stabilization"

_pharmaceutics, 2023, doi:10.3390/pharmaceutics15020620_

Round 1
Reviewer 1 Report
Very well written manuscript.
Nicely summarised. Good suggestions for future work.
It may benefit from a table of "factors affecting mRNA stability" and either in the same table or a separate one with the ways to protect stability.
One minor typo
title 3.3 "manufacturing" not "manufactory"
Author Response
General comment:
Very well written manuscript.
Nicely summarised. Good suggestions for future work.
Response:
We thank Reviewer #1 very much for the positive comment and encouragement on our work. We have considered your two comments and revised our manuscript accordingly.
Comment 1: It may benefit from a table of "factors affecting mRNA stability" and either in the same table or a separate one with the ways to protect stability.
Response 1: We thank Reviewer 1 for this constructive suggestion. A table summarizing “factors affecting mRNA stability”and possible ways to protect mRNA as below has been added to the manuscript (page 3 & 4).
Table 1. Factors leading to mRNA degradation and corresponding stabilization strategies.
|
Factors destabilizing mRNA |
Stabilization strategies |
Reference |
|
RNase mediated |
1) Modification of mRNA regulatory elements, nucleobases, and backbone; 2) Optimization of mRNA sequences; 3) Loading/encapsulation of mRNA on/into nanoparticles. |
1) [4, 5, 11, 13, 18, 27, 45-49] 2) [55-59] 3) [22, 23, 28, 33, 70-80, 82, 84-87] |
|
High temperature |
Freezing or lyophilization of nanoformualtions with the addition of cryoprotectants. |
[16, 29, 30, 90-93] |
|
Oxidative species |
Introducing chemical groups with antioxidant properties. |
[62] |
Comment 2: title 3.3 "manufacturing" not "manufactory"
Response 2: We thank Reviewer 1 for the kind comment. Title 3.3. has been revised to “Other considerations before manufacturing”. (Page 11, line 389)

Reviewer 2 Report
In this Review manuscript, Wang et al. summarized nanobiotechnology-based approaches to stabilize mRNA. Briefly, they first discussed the mRNA degradation pathways and then investigated various ways to improve mRNA stability in the literature, including the use of chemical modifications, and nanomaterials (e.g., polymer/lipid nanoparticles and hydrogels). Finally, the authors discussed mRNA stabilization during manufacturing and offered their perspectives and outlooks. Overall, I think it is an important topic and is worth covering as mRNA has recently emerged as a promising therapeutic modality. Although many reviews have already covered the delivery of mRNA therapeutics, only a few mentioned how to improve mRNA stability. The manuscript is well-written and well-structured, and importantly, it provided a comprehensive summary of relevant advances in the field. Therefore, I recommend publication after minor revisions.
· Line 77, the authors discussed the degradation pathway of mRNA through the 2’OH group. However, I do not think low pH can increase the nucleophilicity of the OH group. The authors should double-check if it is correct.
· Perhaps, I think it would be better if the authors could add a figure for section 3.1.1 where they could summarize the chemical modification approaches.
Author Response
General comment:
In this Review manuscript, Wang et al. summarized nanobiotechnology-based approaches to stabilize mRNA. Briefly, they first discussed the mRNA degradation pathways and then investigated various ways to improve mRNA stability in the literature, including the use of chemical modifications, and nanomaterials (e.g., polymer/lipid nanoparticles and hydrogels). Finally, the authors discussed mRNA stabilization during manufacturing and offered their perspectives and outlooks. Overall, I think it is an important topic and is worth covering as mRNA has recently emerged as a promising therapeutic modality. Although many reviews have already covered the delivery of mRNA therapeutics, only a few mentioned how to improve mRNA stability. The manuscript is well-written and well-structured, and importantly, it provided a comprehensive summary of relevant advances in the field. Therefore, I recommend publication after minor revisions.
Response:
We thank Reviewer #2 very much for the positive comment and encouragement on our work. We have considered your two comments and revised our manuscript accordingly.
Comment 1: Line 77, the authors discussed the degradation pathway of mRNA through the 2’OH group. However, I do not think low pH can increase the nucleophilicity of the OH group. The authors should double-check if it is correct.
Response 1: We thank Reviewer #2 for pointing out this problem. We agree with your comment. Accordingly, the discussion on the effect of low pH on hydroxyl group nucleophilicity and mRNA instability has been removed.
Comment 2: Perhaps, I think it would be better if the authors could add a figure for section 3.1.1 where they could summarize the chemical modification approaches.
Response 2: We thank Reviewer #2 for this useful comment. A attached figure summarizing chemical modification approaches has now been added to section 3.1.1 (as Figure 2 in page 3), the following discussion has also been updated in the revised manuscript.
In the coding region, both the mRNA backbone and nitrogenous bases can be modified. For mRNA backbone modification, the non-bridging oxygen atom can be replaced with sulfur through phosphorothioate modification (Figure 2a). As the S atom is less electronegative than O, mRNA with this modification becomes a bad substrate against RNase-mediated in-line hydrolysis [48]. Due to nucleophilic attack, the phosphorus atom becomes a chiral center that has two possible configurations: Sp and Rp. It is reported that the Sp configuration of the central phosphate atom makes mRNA more stable compared to the Rp counterpart, probably due to reduced RNase recognition [49].
For base modification in the coding region, the replacement of uridine (U) with pseudouridine (ψ) reduces mRNA in-line hydrolysis (Figure 2b), which could be attributed to two reasons. One is that ψ downregulates 2’-5’-oligoadenylate synthetase activity, thus inhibiting the activity of 2'-5' oligoadenylate synthetase-dependent ribonuclease (RNase L) [50,51] and reduces RNase L mediated in-line hydrolysis. The other is that the nucleophilicity of the 2’OH group of the modified nucleotide is reduced [36]. This molecular modification strategy has been successfully applied in some mRNA cancer vaccines and COVID-19 vaccines [52].

Reviewer 3 Report
H. Xian, C. Yu, Y. Wang, and their colleagues reviewed a hot topic entitled “Nanobiotechnology enabled mRNA stabilization.” The authors mainly focused on two topics: factors that cause mRNA degradation during manufacturing, storage, and upon introduction into the cells, and nanobiotechnology-enabled strategies that protect the mRNA from RNase attacks and storage hydrolysis.
The review is well-framed and well-written. However, the authors should have included many essential and elegant strategies that protect the mRNA from hydrolysis. The authors should consider introducing the following suggestions and detailed revisions addressing the following issues carefully before considering it for a possible publication in MDPI Pharmaceutics to reach broad audiences and readers.
1. Line 24: Give the commercial names of mRNA vaccines; Comirnaty® (Pfizer-BioNTech) and Spikevax®(Moderna).
2. Line 50: The authors must discuss how the anionic cellular membrane obstructs the cellular uptake of negatively charged mRNA and then connect this entry barrier with the development of nanoparticles. For example, a negatively charged surface of mRNA prevents interactions with the anionic cellular membrane, thereby hampering the effective internalization of mRNA into the cells.
3. Describe what in-line hydrolysis of mRNA is. How is it different from hydrolysis by ribonucleases, oxidizers, and chemical modifiers? This must be introduced right after the first appearance of in-line hydrolysis. For example, in-line hydrolysis is the cleavage of a backbone phosphodiester bond in mRNA by a 2′ hydroxyl group of the ribose sugar that is in line with the phosphate group and leaves 5′ oxygen (Figure 1B).
4. The authors should also introduce that mRNA degradation by ribonucleases. It is recommended to introduce this critical degradation factor inside cells using proper citations. For example, mRNA is rapidly degraded by exoribonucleases and endoribonucleases present in the extracellular and intracellular spaces even though packaged inside the nanoparticles (J Drug Target. 2019;27(5-6):670-680, Mol Pharm. 2018;15(6):2268-2276, and Biomater Sci. 2022;10(5):1166-1192).
5. The information in lines 77, 78, and 79 seem scientifically misleading: The authors introduced that low pH of less than 2 (e.g., in gastric juice) can increase the nucleophilicity of the 2’OH group, thereby exacerbating intrastrand breakage. If this statement is true, why is commercial mRNA kept in citric acid buffer pH 4.5? Please crosscheck whether this statement is correct or not. I think deprotonation of the 2’OH group occurs in the basic pH.
6. Line 87: Give the reference to line 87: The above events can occur either intracellularly or extracellularly (J Drug Target. 2019;27(5-6):670-680).
7. I recommend that the authors introduce the following strategies that help protect the mRNA from RNase-mediated degradation with proper citations. I welcome the authors to discuss many other strategies that helped mRNA protection in the harsh biological milieu.
Complexation of mRNA using poly(l-ornithine) with a side chain trimethylene spacer enhanced mRNA protection from ribonuclease attacks compared to poly(l-lysine) possessing a tetramethylene spacer(Macromol Rapid Commun. 2022;43(12):e2100754). Furthermore, the coating of negatively charged polymer onto the positively charged binary polyplexes additionally offered mRNA protection, thereby enhancing protein protection (Macromol Rapid Commun. 2022;43(12):e2100754, Pharmaceutics2021;13(1):126).
Packaging mRNA into a non-crosslinked polyplex micelle of PEG-poly(l-lysine) efficiently protected the remaining mRNA by more than 10,000-fold compared to the naked mRNA in the serum conditions. Furthermore, cationic charge-preserved disulfide crosslinking (e.g., 1-amidine-3-mercaptoproyl groups) at the thiolation degree of 13% improved mRNA protection to 4.9-fold compared to non-crosslinked polyplex micelle. Overall, cationic charge-preserved disulfide crosslinking offered protection to mRNA by more than 49,000-fold compared to naked mRNA (J Drug Target 2019;27(5-6):670-680).
Stimuli-responsive RNA oligonucleotides that bridge mRNA and polycation segments of PEG-block copolymers are constructed to further protect the mRNA from RNase attacks (Adv Healthc Mater2022;11(9):e2102016).
The mRNA polyion complexes constructed from odd-numbered aminoethylene repeats substantially protected the mRNA from ribonucleases than those prepared from even-numbered aminoethylene repeats, thereby presenting sustainable protein production (J Am Chem Soc 2014;136(35):12396-405).
Surface coating of mRNA onto a positively-charged liposome (RNA-lipoplex) at a charge ratio [lipid (+):mRNA (−)] of 1.3:2 showed a full resistance to degradation in the mouse serum (Nature2016;534(7607):396-401), thereby effectively transfecting the spleen upon intravenous injection of RNA-lipoplex.
The metabolic stability of single-stranded mRNA was dramatically improved by 3000-fold against RNase A after hybridizing a complementary reverse mRNA with forward mRNA to double-stranded mRNA (Gene Ther 2018;25(7):473-484).
Encapsulation of mRNA with PEG-block-polycations separately conjugated with phenylboronic acid and polyol groups substantially increased the tolerability of mRNA against ribonuclease attacks by forming a crosslinked polyplex micelle via spontaneous phenylboronate ester formation (J Control Release 2021;330:317-328).
Circularization of mRNA (circRNA) lacks 5′ and 3′ free ends necessary for exonuclease-mediated degradation, thereby rendering it resistant to exonucleases and granting it extended half-life compared to its linear mRNA (Nucleic Acids Res 2016;44(3):1370-83). 5% introduction of N6-methyladenosine (m6A)into circRNA showed more resistance against nucleases (10.1038/s41587-022-01393-0).
8. Stereochemical information (cis, E, R, etc.; d, L), locants (N-methyl), symmetry groups and space groups (C2v), and prefixes in formulae or compound names (tBu and tert-butyl) must be in italics. For example, Line 124: O-to-S substitution O-to-S substitution Line 127: N6,2′-O-dimethyladenosine N6,2′-O-dimethyladenosine
9. Better to show some information in italics. block and co. For example, Line 276: poly(amino-co-ester) poly(amino-co-ester) Line 258: poly(β-amino ester) poly(β-amino ester)
10. Line 49: modified chemically modified
11. Reference 21: Change the preprint reference to the recently published Nucleic Acids Research(https://doi.org/10.1093/nar/gkab764)
12. Expand the acronyms right after their first appearance in the text.
For example, Line 197: PEG Poly(ethylene glycol)
Later, it is necessary to use—for example, line 277: polyethylene glycol (PEG)
Author Response
General comment: H. Xian, C. Yu, Y. Wang, and their colleagues reviewed a hot topic entitled “Nanobiotechnology enabled mRNA stabilization.” The authors mainly focused on two topics: factors that cause mRNA degradation during manufacturing, storage, and upon introduction into the cells, and nanobiotechnology-enabled strategies that protect the mRNA from RNase attacks and storage hydrolysis.
The review is well-framed and well-written. However, the authors should have included many essential and elegant strategies that protect the mRNA from hydrolysis. The authors should consider introducing the following suggestions and detailed revisions addressing the following issues carefully before considering it for a possible publication in MDPI Pharmaceutics to reach broad audiences and readers.
Response:
We thank Reviewer #3 for the positive comments and very helpful suggestions on improving the quality of our manuscript. All your valuable comments have been carefully considered and our manuscript is revised accordingly, see details below.
Comment 1: Line 24: Give the commercial names of mRNA vaccines; Comirnaty® (Pfizer-BioNTech) and Spikevax®(Moderna).
Response 1: We thank Reviewer #3 for the helpful comment. The commercial names of these two COVID vaccines have been added in the revised manuscript (see below).
Lipid nanoparticle (LNP) encapsulated-mRNA vaccines produced by Moderna (Spikevax®) [2] and Pfizer-BioNTech (Comirnaty®) [3] have helped millions of people against COVID-19. (line 24-25, page 1)
Comment 2: Line 50: The authors must discuss how the anionic cellular membrane obstructs the cellular uptake of negatively charged mRNA and then connect this entry barrier with the development of nanoparticles.
For example, a negatively charged surface of mRNA prevents interactions with the anionic cellular membrane, thereby hampering the effective internalization of mRNA into the cells.
Response 2: We thank Reviewer #3 for the constructive comment. The following statement has been added in the revised manuscript (line 50-52).
Yet the negatively charged surface of mRNA repulses with the anionic cell membranes, which hampers the effective internalization of mRNA into cells.
Comment 3: Describe what in-line hydrolysis of mRNA is. How is it different from hydrolysis by ribonucleases, oxidizers, and chemical modifiers? This must be introduced right after the first appearance of in-line hydrolysis.
For example, in-line hydrolysis is the cleavage of a backbone phosphodiester bond in mRNA by a 2' hydroxyl group of the ribose sugar that is in line with the phosphate group and leaves 5’ oxygen (Figure 1B).
Response 3: We thank Reviewer #3 for the useful comment. To better describe in-line hydrolysis and distinguish it from ribonuclease-, oxidizer-, and chemical modifier- accelerated hydrolysis, the following text has been added to our revised manuscript (line 76-79).
"In-line hydrolysis is the cleavage of a backbone phosphodiester bond in mRNA by a 2’ hydroxyl group of the ribose sugar that is in line with the phosphate group and leaves 5’ oxygen (detailed reaction mechanism shown in Figure 1b) [16]. "
Comment 4: The authors should also introduce that mRNA degradation by ribonucleases. It is recommended to introduce this critical degradation factor inside cells using proper citations.
For example, mRNA is rapidly degraded by exoribonucleases and endoribonucleases present in the extracellular and intracellular spaces even though packaged inside the nanoparticles (J Drug Target. 2019;27(5-6):670-680, Mol Pharm. 2018;15(6):2268-2276, and Biomater Sci. 2022;10(5):1166-1192).
Response 4: We thank Reviewer #3 for the important comment. Relevant information and citations regarding ribonucleases have been added to section 2 (line 88).
"Particularly, mRNA may be rapidly degraded by exoribonuclease and endoribonuclease present intracellularly and extracellularly even packaged inside the nanoparticles [33-35]."
Comment 5: The information in lines 77, 78, and 79 seem scientifically misleading: The authors introduced that low pH of less than 2 (e.g., in gastric juice) can increase the nucleophilicity of the 2’ OH group, thereby exacerbating intrastrand breakage. If this statement is true, why is commercial mRNA kept in citric acid buffer pH 4.5? Please crosscheck whether this statement is correct or not. I think deprotonation of the 2’OH group occurs in the basic pH.
Response 5: We thank Reviewer #3 for the kind comment. Following your advice, we have removed the related discussion in the revised manuscript.
Comment 6: Line 87: Give the reference to line 87: The above events can occur either intracellularly or extracellularly (J Drug Target. 2019;27(5-6):670-680).
Response 6: We thank Reviewer #3 for the constructive comment. We have now cited this paper as reference [33].
The above events can occur either intracellularly or extracellularly [33].
Comment 7: I recommend that the authors introduce the following strategies that help protect the mRNA from RNase-mediated degradation with proper citations. I welcome the authors to discuss many other strategies that helped mRNA protection in the harsh biological milieu.
Complexation of mRNA using poly(L-ornithine) with a side chain trimethylene spacer enhanced mRNA protection from ribonuclease attacks compared to poly(L-lysine) possessing a tetramethylene spacer (Macromol Rapid Commun. 2022;43(12):e2100754). Furthermore, the coating of negatively charged polymer onto the positively charged binary polyplexes additionally offered mRNA protection, thereby enhancing protein protection (Macromol Rapid Commun. 2022;43(12):e2100754, Pharmaceutics2021;13(1):126).
Packaging mRNA into a non-crosslinked polyplex micelle of PEG-poly(L-lysine) efficiently protected the remaining mRNA by more than 10,000-fold compared to the naked mRNA in the serum conditions. Furthermore, cationic charge-preserved disulfide crosslinking (e.g., 1-amidine-3-mercaptoproyl groups) at the thiolation degree of 13% improved mRNA protection to 4.9-fold compared to non-crosslinked polyplex micelle. Overall, cationic charge-preserved disulfide crosslinking offered protection to mRNA by more than 49,000-fold compared to naked mRNA (J Drug Target 2019;27(5-6):670-680).
Stimuli-responsive RNA oligonucleotides that bridge mRNA and polycation segments of PEG-block copolymers are constructed to further protect the mRNA from RNase attacks (Adv Healthc Mater2022;11(9):e2102016).
The mRNA polyion complexes constructed from odd-numbered aminoethylene repeats substantially protected the mRNA from ribonucleases than those prepared from even-numbered aminoethylene repeats, thereby presenting sustainable protein production (J Am Chem Soc 2014;136(35):12396-405).
Surface coating of mRNA onto a positively-charged liposome (RNA-lipoplex) at a charge ratio [lipid (+):mRNA (−)] of 1.3:2 showed a full resistance to degradation in the mouse serum (Nature2016;534(7607):396-401), thereby effectively transfecting the spleen upon intravenous injection of RNA-lipoplex.
The metabolic stability of single-stranded mRNA was dramatically improved by 3000-fold against RNase A after hybridizing a complementary reverse mRNA with forward mRNA to double-stranded mRNA (Gene Ther 2018;25(7):473-484).
Encapsulation of mRNA with PEG-block-polycations separately conjugated with phenylboronic acid and polyol groups substantially increased the tolerability of mRNA against ribonuclease attacks by forming a crosslinked polyplex micelle via spontaneous phenylboronate ester formation (J Control Release 2021;330:317-328).
Circularization of mRNA (circRNA) lacks 5′ and 3′ free ends necessary for exonuclease-mediated degradation, thereby rendering it resistant to exonucleases and granting it extended half-life compared to its linear mRNA (Nucleic Acids Res 2016;44(3):1370-83). 5% introduction of N6-methyladenosine (m6A) into circRNA showed more resistance against nucleases (10.1038/s41587-022-01393-0).
Response 7: We thank Reviewer 3 for your insightful comment. The following discussions and new references have been added to our revised manuscript.
Positively charged side chains derived from natural amino acids such as poly(L-ornithine) and poly(L-lysine) have also been applied in polymer design. It was found that complexation using poly(L-ornithine) with a trimethylene spacer better enhanced mRNA from RNase attack than poly(L-lysine) possessing a tetramethylene spacer [74]. It was postulated that the difference in one methylene unit affected the ion pairing between mRNA phosphate group and amino groups in the positively charged side chain [74].
Moreover, the polyplex surface charge can be controlled to avoid the high positive charge density. Binary polyplexes (e.g., mRNA + poly(L-ornithine) mentioned above) can be further coated with negatively charged polymers to offer mRNA protection against serum [74, 75].
(Section 3.2.2 line 287-298)
Furthermore, functional chemical motifs can be introduced to above co-polymer systems to enhance mRNA stability. For example, encapsulating mRNA into a non-crosslinked polyplex micelle of PEG-poly(L-lysine) efficiently protected the mRNA by more than 10,000-fold compared to naked mRNA in serum [33]. Besides, cationic charge-preserved disulfide crosslinking (e.g., 1-amidine-3-mercaptoproyl group) improved mRNA stability to 4.9-fold compared to non-crosslinked polyplex micelle [33].
(Section 3.2.2 line 317-322)
RNA oligonucleotides were also fabricated to link mRNA and polycation segment of PEG-block copolymers, driving the formation of polyplex micelle, further protecting mRNA from RNase attack [81].
(Section 3.2.2 line 323-325)
Uchida and coworkers reported that mRNA condensed from odd-numbered aminoethylene repeats remarkably protected mRNA from RNase degradation than even-numbered ones, leading to enhanced protein production [83]. The authors explained it could be due to difference in the protonation of terminal primary side chain amino groups (pH 7.4), but the exact biochemical mechanism remains to be explored further[83].
(Section 3.2.2 line 332-337)
Another strategy is to hybridize complementary mRNA strands for enhanced mRNA stability. Compared to single-stranded mRNA, the stability of hybridized mRNA improved by 3000-fold against RNase A degradation [58]
(Section 3.1.2 line 190-193)
Encapsulation of mRNA with two PEG-polycation block copolymers conjugated with phenylboronic acid or polyol groups allows spontaneous phenyl-boronate ester formation, driving the formation of a crosslinked polyplex micelle. This significantly improved the resistance of mRNA against RNase [75].
(Section 3.2.2. line 325-328)
In addition, removal of the 5’ and 3’ ends of mRNA that are necessary for exoribonuclease-mediated degradation forms circular RNA (circRNA), prolonging its half-life compared to linear RNA [48]. Moreover, incorporation of 5 % N6-Methyladenosine (m6A) into circRNA further enhanced its resistance to nucleases [49].
(Section 3.1.1. line 141-144)
Comment 8: Stereochemical information (cis, E, R, etc.; d, L), locants (N-methyl), symmetry groups and space groups (C2v), and prefixes in formulae or compound names (tBu and tert-butyl) must be in italics.
For example, Line 131: O-to-S substitutionà O-to-S substitution
Line 134: N6,2′-O-dimethyladenosine à N6,2′-O-dimethyladenosine
Response 8: We thank Reviewer #3 for this suggestion. We have checked this throughout our manuscript and now italicized relevant text (line 132-135).
The standard 5’ cap can be replaced with analogs containing O-to-S substitution within the 5’ phosphoester. Due to this structural change, the recognition of mRNA caps by DCPs (e.g., Dcp2 and DcpS) is reduced, making mRNA more stable [45]. In the 5’ UTR, the first adenosine base can be modified to N6,2’-O-dimethyladenosine (m6Am), a large group that sterically blocks DCP access to the 5’ cap [18].
Comment 9: Better to show some information in italics. block and co. For example, Line 276: poly(amino-co-ester) à poly(amino-co-ester)
Line 258: poly(β-amino ester) à poly(β-amino ester)
Response 9: We thank Reviewer #3 for this suggestion. We have checked this throughout our manuscript and now italicized relevant text (e.g., poly(amino-co-ester) and poly(β-amino ester)).
Line 283: One way to resolve above issue is to design branched polymers with lower charge density compared to PEI, e.g., single branch poly(β-amino ester) (PBAE).
Line 310: Examples of cationic polymers are PEI [79] and poly(amino-co-ester)s (PACEs) [80].
Comment 10: Line 49: modified –> chemically modified
Response 10: We thank Reviewer #3 for this suggestion. The original manuscript has been revised to (line 49): To mitigate mRNA degradation, mRNA molecules can be chemically modified or self-assembled into nanostructures to reduce in-line hydrolysis [18]
Comment 11: Reference 21: Change the preprint reference to the recently published Nucleic Acids Research (https://doi.org/10.1093/nar/gkab764)
Response 11: We thank Reviewer #3 for this kind suggestion. This should be Ref 56 (before revision) in our manuscript – we have now changed it to the recently published work (see below).
- Wayment-Steele, H.K.; Kim, D.S.; Choe, C.A.; Nicol, J.J.; Wellington-Oguri, R.; Watkins, A.M.; Parra Sperberg, R.A.; Huang, P.S.; Participants, E.; Das, R. Theoretical basis for stabilizing messenger RNA through secondary structure design. Nucleic Acids Res 2021, 49, 10604-10617, doi:10.1093/nar/gkab764.
Comment 12: Expand the acronyms right after their first appearance in the text.
For example, Line 197: PEG -> Poly(ethylene glycol)
Later, it is necessary to use—for example, line 277: polyethylene glycol (PEG)
Response 12: We thank Reviewer #3 for this kind comment. All acronyms in this manuscript have been checked again and revised accordingly.
Line 217: LNPs are usually composed of four components: cationic/ionizable lipids, helper lipids (e.g., phospholipids), poly(ethylene glycol)-modified lipids (i.e., PEGylated lipids), and cholesterol [1, 44, 57]
Line 314/315: An example of a neutral polymer is PEG.

Round 2
Reviewer 3 Report
1. Lines 300, 303, and 309: The configurational descriptors [d-, l-, and dl-] must be in small capitals. Not italics and big capitals. You can find more information from the following link. https://www.cas.org/sites/default/files/documents/indexguideapp.pdf
poly(L-ornithine) and poly(L-lysine) poly(l-ornithine) and poly(l-lysine)
Type l or d Go to Format Go to Font Select small caps shown in “effects”.
2. Line 93: The authors mentioned that exoribonuclease and endoribonuclease may even rapidly degrade nanoparticle-packaged mRNA intracellularly and extracellularly. This sentence is scientifically misleading. Packaging mRNA inside the nanocarriers effectively protected against RNAase attacks compared to naked mRNA. However, nanocarrier-encapsulated mRNA is also subjected to RNase attacks. It may be better to remove “rapidly”, as nanoparticles protect the mRNA relatively longer time compared to naked mRNA.
Author Response
Comment 1:
- Lines 300, 303, and 309: The configurational descriptors [d-, l-, and dl-] must be in small capitals. Not italics and big capitals. You can find more information from the following link. https://www.cas.org/sites/default/files/documents/indexguideapp.pdf
poly(L-ornithine) and poly(L-lysine) poly(l-ornithine) and poly(l-lysine)
Type l or d Go to Format Go to Font Select small caps shown in “effects”.
Response 1:
We thank reviewer for this kind comment. We have corrected this in the revised draft (line 287-296).
Polymers with positively charged side chains derived from natural amino acids such as poly(l-ornithine) and poly(l-lysine) have also been applied in polymer design. It was found that complexation using poly(l-ornithine) with a trimethylene spacer better enhanced mRNA from RNase attack than poly(l-lysine) possessing a tetramethylene spacer [74]. It was postulated that the difference in one methylene unit difference affected the ion pairing between mRNA phosphate group and amino groups in the positively charged side chain [74].
Moreover, the polyplex surface charge can be controlled to avoid the high positive charge density. Binary polyplexes (e.g., mRNA + poly(l-ornithine) mentioned above) can be further coated with negatively charged polymers to offer mRNA protection against serum [74,75].
Comment 2:
Line 93: The authors mentioned that exoribonuclease and endoribonuclease may even rapidly degrade nanoparticle-packaged mRNA intracellularly and extracellularly. This sentence is scientifically misleading. Packaging mRNA inside the nanocarriers effectively protected against RNAase attacks compared to naked mRNA. However, nanocarrier-encapsulated mRNA is also subjected to RNase attacks. It may be better to remove “rapidly”, as nanoparticles protect the mRNA relatively longer time compared to naked mRNA.
Response:
We thank reviewer for this kind comment. The word ‘rapidly’ has been removed (line 89).
